# Development and Validation of Flaxseed Lignan-Enriched Set-Type Fermented Milk to Manage Postmenopausal Osteoporosis

Elizabeth Thomas [1], Narender Raju Panjagari [1,*] , Sangita Ganguly [1,*], Sameni Deepika [2], Suman Kapila [2] and Ashish Kumar Singh [1]

1 Dairy Technology Division, National Dairy Research Institute, Karnal 132001, Haryana, India; elizabethblesson1109@gmail.com (E.T.); aksndri@gmail.com (A.K.S.)
2 Animal Biochemistry Division, National Dairy Research Institute, Karnal 132001, Haryana, India; venugopalchinni14@gmail.com (S.D.); skapila69@gmail.com (S.K.)
* Correspondence: narender.p@icar.gov.in (N.R.P.); sangitandri@gmail.com (S.G.)

**Abstract:** A functional set *dahi* (fermented milk analogous to yoghurt) with a desirable probiotic (*Lactiplantibacillus plantarum* A5) count of 9.36 log CFU/mL and excellent techno-functional attributes (DPPH: 41.95% RSA, firmness: 485.49 g, sensory overall acceptability: 8.51) was developed to contain 260 mg of SDG in 20 g of *dahi*. Twenty-four female Albino Wistar rats (3 months old, >180 g) were ovariectomized (OVX) and divided into three groups: OVX control, OVX and control *dahi*, and OVX and SDG-enriched *dahi*. The animal study found that ovariectomy decreased serum calcium, oestrogen, and bone ash calcium levels by 32.27, 30.95, and 48.46 percent, respectively, compared to the sham group (*n* = 8), while daily administration of SDG-enriched *dahi* (20 g) for eight weeks restored them. The proximal tibial metaphysis and distal femoral epiphysis micro-CT study showed that the ovariectomy lowered bone mineral density (BMD) by 11.06% and 9.18%, respectively, and lowered Trabecular thickness (Tb. Th) by 12.66% and 11.86%, respectively, while increasing Trabecular separation (Tb. Sp.) by 90.69% and 87.70%, respectively, compared to the sham control-group rats. SDG-enriched *dahi* improved BMD by 16.06 and 12.24% and Tb. Th by 35.32 and 19.62%, respectively, and decreased Tb. Sp by 47.04 and 47.22%, respectively, in OVX rats. The results suggest that the developed set *dahi* may help treat postmenopausal osteoporosis.

**Keywords:** lignan; *dahi*; postmenopausal osteoporosis; bone mineral density; trabecular microarchitecture

## 1. Introduction

Flaxseed-based foods play a fundamental role in disease prevention as they are abundant in polyunsaturated fatty acids (α-linolenic acid), soluble and insoluble (dietary) fibre, and the plant lignan secoisolariciresinol diglucoside (SDG), which may all help with disease prevention and health promotion [1]. SDG is a phytoestrogen, or plant hormone, and flaxseed has been determined to be the most significant source of SDG among diverse plant foods, with almost 1000 times the amount of SDG found in sesame seeds, pumpkin seeds, wheat, lentils, soybeans, pears, prunes, garlic, asparagus, and carrots [2]. SDG levels in defatted flaxseed powder have been reported to vary between 6 and 29 g/kg [3]. Upon ingestion, gut bacteria break down SDG into enterolignans, namely enterodiol (END) and enterolactone (ENL). Several studies have highlighted the potential effects of flaxseed lignan and its metabolites, enterodiol and enterolactone, on mammalian and prostate cancer, menopausal symptoms, cholesterol, obesity, hypertension, osteoporosis, diabetes, and cardiovascular diseases [4].

Osteoporosis is a skeletal disease characterized by reduced bone mineral density (BMD), the weakening of bones, and increased fracture risk. The World Health Organization describes osteoporosis as a significant health issue as hip fractures due to the disease

are expected to rise from 4.5 to 6.3 million by 2050 [5]. After menopause, the sudden decline in oestrogen levels causes postmenopausal osteoporosis since oestrogen is primarily responsible for bone remodelling and bone stability. In men and women, oestrogen slows the remodelling rate while concurrently controlling the activity of osteoblasts and osteoclasts to maintain bone formation and prevent bone resorption. The enterolignans END and ENL are structurally similar to human oestrogen, and they exhibit binding affinity, although low, to oestrogen receptors α and β. There needs to be more research on the potential benefits of SDG on bone metabolism, particularly in preventing or treating postmenopausal osteoporosis. Earlier studies have reported conflicting results regarding bone health. According to [6], a moderate dose of SDG in neonates resulted in denser bones at maturity, suggesting that dietary SDG may improve bone development. However, flaxseed supplementation did not change biochemical indicators of bone metabolism or BMD in healthy postmenopausal women in short-term (3 months) and long-term (12 months), randomized, double-blind, placebo-controlled trials [7,8]. Also, it was revealed that SDG consumption throughout growth negatively affected weight and bone mineral content (BMC) in female rats [9]. Some previous clinical studies have administered SDG directly or with the basal diet to evaluate its effect on postmenopausal women. However, none have examined the enrichment of fermented milk products and their effect on postmenopausal osteoporosis.

Recently, reports have shown that certain strains of Lactobacilli can maintain bone mass in oestrogen-deficient rodents and humans [10–13]. The administration of *Lactiplantibacillus plantarum* MGE 3038 to OVX rats (12-week-old female Wistar rats) for 16 weeks showed enhanced maintenance of trabecular bone structure [14]. Similarly, *Lactiplantibacillus plantarum* GMNL-662 was proved to inhibit ovariectomy-induced osteoporosis through the modulation of proinflammatory cytokines and bone metabolism-related markers [10]. In another work, the administration of *Lactobacillus acidophilus* in OVX mice was shown to enhance both trabecular and cortical bone microarchitecture along with increasing the mineral density and heterogeneity of bones [15]. Similarly, *Lactobacillus reuteri* ATCC PTA 6475 has proved to be effective in treating menopausal osteoporosis in mouse models [16]. These studies suggest that certain probiotic strains of *Lactobacillus* increase the bioavailability of calcium by producing short-chain fatty acids that lower gut pH, making it easier for calcium to dissolve and be absorbed, and, further, they exhibit immunomodulatory effects on the host immune system [17]. In addition, fermented dairy products help to maintain bone homeostasis as they are abundant in calcium, phosphate, micronutrients, and protein. Enriching SDG in fermented milk in the presence of a probiotic bacteria (*Lactiplantibacillus plantarum* A5) may lead to developing a functional fermented dairy food to alleviate postmenopausal osteoporosis. Hence, the subsequent aim of the investigation was to assess the impact of flaxseed lignan-enriched set *dahi* on postmenopausal osteoporosis, utilizing the established OVX rat model approved by the Food and Drug Administration.

## 2. Materials and Methods

### 2.1. Materials

The flaxseed cultivar used in the study was acquired in bulk (40 kg) from a local food store (Sona Superstore, Karnal, Haryana, India). The whole buffalo milk used in the study was sourced from the Experimental Dairy of National Dairy Research Institute (NDRI), Karnal. Starter culture *Streptococcus salivarius* ssp. *thermophilus* (NCDC 74) was obtained from the NCDC (National Collection of Dairy Cultures) at NDRI, Karnal (India). The glycerol stocks (40% glycerol maintained at −80 °C) of probiotic isolates *Lactiplantibacillus plantarum* A5, previously characterized for probiotic attributes [18], were collected from the Molecular Biology Unit of the Dairy Microbiology Division of NDRI, Karnal. LIQUIXX-M Alkaline Phosphatase Reagent (6 mL × 6) for in vitro diagnostic use was obtained from Transasia Bio-Medicals Ltd., Mahakuma Namchi, South Sikkim, India. Rat Osteocalcin ELISA Kit (Cat: ELK2391) was purchased from ELK Biotechnology, Co., Ltd., Wuhan, China.

*2.2. Development of Flaxseed Lignan-Enriched Set Dahi*

Our previous research describes the extraction of SDG by direct alkaline hydrolysis coupled with magnetic stirring, microwave, and ultrasound; HPLC and FTIR estimation of SDG from defatted whole flaxseed and hull; and their effect on starter and probiotic culture [19]. Briefly, SDG was extracted from milled defatted flaxseed flour using direct alkaline hydrolysis coupled with magnetic stirring. SDG in the extract was measured to be 13 mg/g flaxseed (dry matter basis). The flaxseed extract was rotary-evaporated at 60 °C. Previous studies on extraction of SDG using direct alkaline hydrolysis have reported that SDG obtained through this method is non-toxic as this process destructs the cyanogenic glycosides, producing an extract free of cyanogenic glycosides or free cyanide [20]. The concentrate was then added to low-fat milk (1.5% fat and 11% solids-not-fat (SNFs)) such that the concentration of SDG in 20 g *dahi* was 260 mg (it was found during trial that 20 g is the maximum quantity of *dahi* which can be fed to a rat per day). The SDG-enriched milk was homogenized (2000/500 psi) and heat-treated at 90 °C/10 min. After cooling to 37 °C, 2% of the NCDC 74/A5 (*Streptococcus salivarius* ssp. *thermophilus*/*Lactiplantibacillus plantarum* A5) co-culture was added and incubated at 37 °C for 7 h.

*2.3. Analysis of the SDG-Enriched Dahi Samples*

2.3.1. Proximate Composition Analysis

Total solids, fat, protein, and ash were measured using the methods described by FSSAI in "Manual of Methods of Analysis of Foods (Milk and Milk Products)" [21].

2.3.2. Titratable Acidity

Titratable acidity of the *dahi* was evaluated according to the standard protocol of Association of Official Agricultural Chemists [22].

2.3.3. Determination of pH

The pH of the samples was measured using a digital pH meter (EUTECH Instruments, Singapore) at 20 °C using a combined glass electrode fitted in association with a temperature probe. Before use, the pH meter was calibrated using standard buffers of pH 4.0 and 9.0 at 20 °C. Triplicate measurements were taken for each sample to determine pH.

2.3.4. Antioxidant Activity (DPPH)

DDPH radical scavenging activity (RSA) was analysed using a previously described protocol [23]. Briefly, 1 g of curd sample was diluted in 25 mL of 60% methanol. Further, 2.9 mL of 60 mM DPPH in methanol was added to 100 μL of the diluted sample and kept in the dark for 30 min. The absorbance of the reaction mixture was then measured at 517 nm. As a blank, methanol was utilized. Triplicate measurements were taken. RSA was calculated by using the following formula:

$$\text{RSA}(\%) = \frac{\left( \text{Absorbance}_{\text{blank}} - \text{Absorbance}_{\text{sample}} \right) \times 100}{\text{Absorbance}_{\text{blank}}}$$

2.3.5. Textural Attributes

Textural qualities were measured using an TA-XT plus texture analyser (M/s Stable Micro Systems, Godalming, UK) with a 50 kg load cell and back extrusion method. Inoculated milk (~140 mL) was filled to 5 cm in a pre-sterilized glass tumbler (10 cm height and 6 cm diameter) and incubated to assess texture. The beakers were tempered for 2 h at 25 °C before analysis. A back extrusion probe (A/BE) with a 5 cm disc diameter and extension bar was used. The extrusion disc was positioned centrally over the sample container. At a crosshead speed of 1.0 mm/s, it penetrated 10 mm (20% compression) into the set *dahi*. The probe compressed and back-extruded the material, raising fluid in the annular space. The instrument-attached software (Exponent Connect$^{\text{TM}}$, Version 7,0,6,0) calculated firmness (g), consistency (g·s), cohesiveness (g), and work of cohesion (g·s) from the force–time

curves. Triplicate measurements were taken for each sample. The parameters and the TA settings used are provided in the Supplementary File.

### 2.3.6. Sensory Analysis

A sensory review panel of ten judges was chosen from the Dairy Technology Division of NDRI, Karnal, who were well-versed in sensory evaluation procedures and product qualities. The sensory qualities of set *dahi* were evaluated in the study, including colour and appearance, flavour, body and texture, and overall acceptance. The sensory evaluation tool used a nine-category hedonic grading system: dislike extremely (1), dislike very much (2), dislike moderately (3), slightly dislike (4), neither like nor dislike (5), slightly like (6), moderately like (7), like very much (8), and extremely like (9).

### 2.3.7. Total Viable Bacterial Count

Plate count agar was used for the enumeration of total viable bacteria count using the standard method. The 7th, 8th, and 9th dilutions were used with the spread plate technique for enumeration. Plating was conducted in triplicate and the incubation was performed at 37 °C for 24 to 48 h. The obtained results were expressed in terms of the $Log_{10}$ colony forming units ($Log_{10}$ CFUs/mL) of the sample.

### 2.3.8. Probiotic Count

For the selective enumeration of probiotic *L. plantarum*, MRS-C (de Man, Rogosa, and Sharpe) medium agar supplemented with ciprofloxacin (5 µg/mL) was used. The 5th, 6th, and 7th dilutions were used with the spread plate technique for enumeration. Plating was conducted in triplicate and the incubation was performed at 37 °C for 24 to 48 h. The obtained results were expressed in terms of the $Log_{10}$ CFU/mL of the sample.

### *2.4. Establishment of Osteoporotic Rat Model by OVX (Experimental Design and Treatments)*

The present study was conducted with approval from the Institutional Animal Ethics Committee (IAEC), NDRI, Karnal, India. Thirty-two female Albino Wistar rats, weighing 180 ± 10 g and aged three months, were subjected to sham surgery (*n* = 8) or surgical ovariectomy (*n* = 32) under anaesthesia induced by intramuscular injection of 50 mg/kg ketamine and 12.5 mg/kg xylazine in a total volume of 320 µL through an intraperitoneal route [24]. Bilateral ovariectomy (OVX) was performed by dorsal approach to remove both ovaries. Sham surgery followed the same steps, except ovaries were identified but not removed. Rats were allowed to recover for two weeks before therapy. After recovery, rats were randomly assigned into four groups of eight rats each: Sham, OVX, OVX and control *dahi*, and OVX and SDG-enriched *dahi*. The sham and OVX groups received standard rat chow and water ad libitum for eight weeks. The OVX and control *dahi* group received a 40 g mixed meal (rat chow and control *dahi* mixed in 1:1 ratio), while the OVX and SDG-enriched *dahi* group received a 40 g mixed diet (20 g rat chow and 20 g SDG-enriched *dahi*, ~260 mg SDG/rat per day) and ad libitum water. The changes in the body weights of all the rats were recorded on day 0 and then weekly for the entire 8-week duration of the study. A total of 24 h after the last treatment, rats were fasted for 16 h. Rats were anesthetized, sacrificed, and necropsied.

After administering mild anaesthesia to the animals and drawing blood from the heart via cardiac puncture, the blood was centrifuged at 4000 rpm for 10 min at 4 °C in a refrigerated centrifuge. The obtained serum was stored in aliquots at −20 °C until analysis. The femur and tibia from both sides were dissected and cleaned of adhering muscles and soft tissues during the necropsy. The left femur and tibia were fixed for 72 h in 10% neutral buffered formalin (NBF) before being stored in 70% isopropanol until micro-CT analysis. At the same time, the right femur and tibia were dried in a hot air oven at 95 °C for 10 h. The length and dry weight of the right tibia and femur were measured prior to determining the calcium content of bone ash using an atomic absorption spectrometer (AAS).

## 2.5. Measurement of Serum Calcium and Phosphorus Level

The serum calcium concentration was determined using the calcium arsenazo III colorimetric procedure. The ammonium molybdate method was used to calculate the serum phosphorus level. The procedure followed can be found in the Supplementary File.

## 2.6. Measurement of Serum Oestradiol

An oestradiol ELISA Kit (M/s Sigma-Aldrich Chemicals Pvt. Ltd., Saint Louis, MO, USA) was used to determine the serum oestradiol concentration according to the manufacturer's protocol. Initially, the chosen coated wells were secured in the holder, and 25 μL of standards, samples, and controls were dispensed into the corresponding wells. After that, 100 μL of the 1× working solution of oestradiol enzyme conjugate was dispensed into each well. It was incubated at room temperature (18–25 °C) for 60 min after being thoroughly mixed for 10–20 s. After incubation, all wells were emptied and washed thrice with 300 μL of 1× wash buffer. Using absorbent paper or paper towel, the wells were then properly blotted. Thereafter, 100 μL of TMB Reagent was added to each well, mixed gently for 10 s, and incubated for 30 min at room temperature (18–25 °C). Further, 50 μL of the stop reagent was applied to each well. The liquid was mixed by tapping the plate. Within 15 min, the microplate reader identified the OD at 450 nm.

## 2.7. Measurement of Ash Weight and Bone Ash Calcium Using AAS

Bone ash calcium levels of the right femur and tibia were measured using the AOAC (2005) technique described in [22]. The right femur and tibia were dried in an incubator at 85 °C for eight hours. Dried femurs and tibiae lengths were measured with a Digital Vernier Calliper (Mitutoyo, India). After weighing the bones in a silica crucible, they were moved to a muffle furnace (Modern Industrial Corporation, Mumbai, India) for 12 h at 550 °C. The crucibles containing the ash of femurs and tibias were removed from the muffle furnace and weighed again. The weight of ash was determined as follows:

$$\text{Ash}(\%) = \frac{\text{Weight of crucible containing ash} - \text{Empty weight of crucible}}{\text{Weight of crucible containing sample} - \text{Empty weight of crucible}} \times 100$$

Calcium from bone ash was solubilized with 10 mL of triple acids (nitric acid/perchloroacetic acid/sulphuric acid) in a ratio of 3:2:1. The ash was then evaporated using a hot plate. The samples were then transferred to 100 mL volumetric flasks, and the volume was brought to 100 mL using milli-Q water. From 100 mL, 200 μL was taken and again diluted to 10 mL using milli-Q water and was used to measure the bone ash calcium level in femurs and tibias with an Atomic Absorption Spectrophotometer (AA-7000, Shimadzu, Kyoto, Japan) at 422.7 nm. After calibrating the instrument with standard calcium carbonate, the calcium concentration in bone ash samples was calculated.

## 2.8. Assessment of BMD and Trabecular Microarchitecture

A Sky Scan 1176 micro-CT scanner (Milabs B.V., Heidelberglaan, Utrecht, The Netherlands) was used to perform micro-CT studies on dissected bones. At the end of eight weeks of treatment, rats were euthanized. Left femurs and tibias were dissected, cleaned of adhering soft tissues, and preserved in 10% neutral buffered formalin for 72 h prior to storage in 70% isopropanol and storage at 4 °C until analysis. The samples were scanned in three batches at a nominal resolution (pixels) of 18 μm. Reconstruction was performed using a modified Feldkamp algorithm using the Sky Scan Nrecon program, which enables network-distributed reconstruction on four concurrently operating personal computers. The X-ray source was calibrated at 70 kV and 100 mA, with an 18 μm pixel size. One hundred projections were collected throughout an angular range of 180°. The picture slices were reconstructed using the cone-beam reconstruction software version 2.6 based on the Feldkamp technique (Skyscan, Edinburgh, UK). BMD of the femoral and tibial bones, Bone Volume/Tissue Volume ratio (BV/TV ratio), Trabecular number (Tb. N), Trabecular thickness (Tb. Th) and Trabecular separation (Tb. Sp) were measured.

*2.9. Measurement of Bone Turnover Marker*

A rat sandwich ELISA assay kit (ELK Biotechnology, Co., Ltd., Wuhan, China) assessed serum osteocalcin (OCN) following the manufacturer's instructions. Standard curve used for the estimation of OCN is provided as supplementary data.

*2.10. Measurement of Bone Formation Marker*

Alkaline phosphatase (ALP) activity was determined with colorimetry using LIQUIXX-M ALP Assay kit (Transasia Bio-Medicals Ltd., Mahakuma Namchi, South Sikkim, India). The method uses 4-nitrophenyl phosphate (NPP) as the substrate. In total, 20 μL serum was mixed with 1000 μL of the reagent containing NPP substrate in cuvette and the absorbance was measured immediately at 405 nm. Then, the mixture was incubated at room temperature for 1 min and again the absorbance was measured at 405 nm. The ALP activity in IU/L can be calculated as follows:

$$\text{ALP activity}\left(\frac{\text{IU}}{\text{L}}\right) = \frac{\Delta\text{A}}{\text{min}}.\text{X Factor}(2764)$$

*2.11. Statistical Analysis*

Data obtained from various experiments were recorded as mean $\pm$ standard deviation (SD) and subjected to statistical analysis. For data analysis, three observations from physico-chemical, textural, and microbial studies; ten observations from sensory evaluation; and six observations from each group of animal study were used. Data were analysed using one-way ANOVA at a 5% level of significance. The mean values were compared using Duncan's paired comparison test using SPSS software (version 26, M/s IBM Corporation, Armonk, NY, USA).

## 3. Results and Discussion

### 3.1. Proximate Composition of SDG-Enriched Set Dahi

Table 1 presents the proximate chemical composition of SDG-enriched set *dahi*. On average, the SDG-enriched set *dahi* contained 13.05% total solids, 1.575% fat, 4.91% protein, 5.55% lactose, and 1.01% ash.

**Table 1.** Proximate chemical composition and physico-chemical, textural, sensory, and microbiological attributes of SDG-enriched set *dahi*.

| Parameters | Control | SDG-Enriched Set *Dahi* |
|---|---|---|
| Total solids (%) | 12.70 $\pm$ 0.14 [a] | 13.05 $\pm$ 0.07 [a] |
| Fat (%) | 1.58 $\pm$ 0.01 [a] | 1.57 $\pm$ 0.01 [a] |
| Protein (%) | 4.75 $\pm$ 0.08 [a] | 4.91 $\pm$ 0.01 [a] |
| Lactose (%) * | 5.57 $\pm$ 0.22 [a] | 5.55 $\pm$ 0.01 [a] |
| Ash (%) | 0.80 $\pm$ 0.01 [a] | 1.01 $\pm$ 0.03 [b] |
| pH | 4.80 $\pm$ 0.02 [b] | 4.67 $\pm$ 0.07 [a] |
| Titratable acidity (%LA) | 0.86 $\pm$ 0.07 [a] | 0.89 $\pm$ 0.08 [a] |
| DPPH (% RSA) | 23.75 $\pm$ 1.10 [a] | 44.13 $\pm$ 0.37 [b] |
| Firmness (g) | 369.39 $\pm$ 10.03 [a] | 323.25 $\pm$ 59.72 [a] |
| Consistency (g·s) | 881.51 $\pm$ 13.45 [b] | 805.38 $\pm$ 27.07 [a] |
| Cohesiveness (g) | $-57.54 \pm 4.42$ [a] | $-262.48 \pm 44.68$ [b] |
| Work of cohesion (g·s) | $-9.03 \pm 0.60$ [a] | $-165.22 \pm 22.37$ [b] |
| Colour and appearance | 8.75 $\pm$ 0.25 [a] | 8.58 $\pm$ 0.52 [a] |
| Body and texture | 8.67 $\pm$ 0.29 [a] | 8.08 $\pm$ 0.38 [a] |
| Flavour | 8.58 $\pm$ 0.14 [a] | 8.00 $\pm$ 0.50 [a] |
| Overall acceptability | 8.42 $\pm$ 0.14 [a] | 8.42 $\pm$ 0.14 [a] |
| TPC (Log$_{10}$CFU/mL) | 10.21 $\pm$ 0.78 [a] | 10.04 $\pm$ 0.42 [a] |
| Probiotic (Log$_{10}$CFU/mL) | 9.36 $\pm$ 0.41 [a] | 9.22 $\pm$ 0.27 [a] |

Control: low-fat (1.5% fat and 11% SNF) set *dahi* formulated with 2% starter culture (NCDC 74/A5) and incubated for 7 h without SDG. [ab] means values with similar superscripts between control and SDG-enriched samples do not differ significantly ($p > 0.05$). Mean $\pm$ S.D. ($n = 3$) except for sensory ($n = 5$). *: on difference basis.

*3.2. Physico-Chemical, Textural, Sensory, and Microbiological Attributes of SDG-Enriched Set Dahi*

The SDG-enriched *dahi* sample's physico-chemical, textural, sensory, and microbiological properties compared to those of the control are shown in Table 1. SDG incorporation significantly ($p \leq 0.05$) impacted the pH, DPPH, and textural aspects of set *dahi*. The addition of SDG significantly enhanced the antioxidant activity ($p \leq 0.05$). Our results correlate with earlier findings that flaxseeds are a good source of antioxidants [25]. Flaxseed varieties exhibited antioxidant activity ranging from 3.25 to 8.40 µmol Trolox equivalent/g. Flaxseed is high in niacin and vitamin E, especially the antioxidant tocopherol. Tocopherol in flaxseed averages 39.5–50 mg/100 g. Also, flaxseed cyclolinopeptides exhibit antioxidant activity [26]. It was also observed that flaxseed powder substantially increases yoghurt RSA. The control yoghurt had 20% RSA, while yoghurt with 5% flaxseed powder added had 48% RSA [27].

The control and set *dahis* with SDG had different textural qualities. SDG-enriched set *dahi* had lower firmness than the control, although the difference was not significant ($p > 0.05$). A significant difference ($p \leq 0.05$) was found in the consistency of the SDG-enriched sample. The consistency decreased by 8.64 percent. The slight rise in moisture content may have disrupted the gel structure and caused whey separation, resulting in an inconsistent hardness and consistency of set *dahi* after adding flaxseed extract. Similar observations have been reported that the addition of aloe vera extract decreases the firmness of *dahi* [28]. The SDG-enriched *dahi* had significantly greater cohesiveness and work cohesion ($p \leq 0.05$) than the control. Previous studies have reported increases in the cohesiveness of *dahi* when supplemented with flaxseed powder, and the increase in cohesiveness is directly related to flaxseed concentration and storage time [29].

Although the scores were low, the sensory qualities of the SDG-enriched set *dahi* and the control were not significantly different ($p > 0.05$). The decrease in sensory scores may be due to the increased moisture content, slight saltiness, and light brown discoloration of the product due to the incorporation of flaxseed lignan extract. Recent research has shown that adding flaxseed to fermented milk decreases sensory scores, but adding flaxseed extract (1–3%) to *kefir* did not affect sensory scores, which was consistent with our findings [30]. Another study discovered no sensory differences between flaxseed lignan-supplemented *misti dahi* and the control [31]. Thus, adding flaxseed as whole flaxseed powder or oil negatively affects dairy products' sensory qualities, while adding flaxseed extract does not. The total viable count and probiotic count of SDG-enriched *dahi* were not significantly different from those of the control ($p > 0.05$). A thorough investigation of the effects of SDG on *Lactiplantibacillus plantarum* and starter cultures found that flaxseed lignan did not impact their fermentation dynamics [19].

*3.3. Effect of SDG-Enriched Set Dahi Administration on Body Weight in OVX Rats*

Figure 1 presents the impact of the administration of control *dahi* and the optimized product on the body weight gain of ovariectomized rats. The mean body weight of the experimental rats did not differ significantly ($p > 0.05$) at the beginning of the study. A statistically significant difference ($p \leq 0.05$) was found between the OVX and OVX and SDG-enriched *dahi* groups in the second week. A significant weight gain difference ($p \leq 0.05$) was seen in the OVX group compared to other groups starting from week 5. A significant difference ($p \leq 0.05$) in weight gain was seen in OVX and the group fed control *dahi* compared to the sham control and OVX and SDG-enriched *dahi* groups in the seventh week of the trial. Compared to untreated OVX rats, daily SDG-enriched *dahi* treatment for eight weeks dramatically reduced body weight gain after ovariectomy. Authors of ovariectomy studies reported similar outcomes. The response of ovariectomized rats to Casein-derived antioxidant peptide (PEP) was studied [24]. They found an increase in body weight after ovariectomy, whereas daily PEP administration at 50 and 100 mg/kg for eight weeks partially prevented body weight gain. Other researchers found that ovariectomized rats gained weight after 5 and 4 weeks [32,33]. In another study, feeding grounded flaxseed

(30 g/kg diet) directly to female rats with induced bone loss regulated body weight gain compared to the untreated group. The flaxseed-treated group gained 49.45%, while the untreated group gained 56.09% [34].

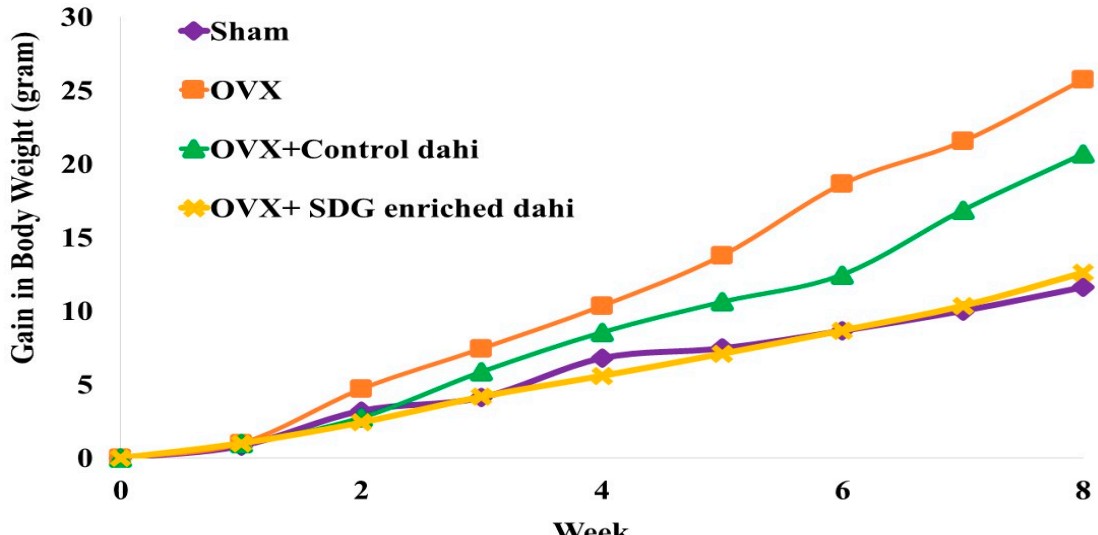

**Figure 1.** Effect of SDG-enriched set *dahi* administration on body weight in OVX rats.

### 3.4. Effect of SDG-Enriched Set Dahi Administration on Serum Calcium, Phosphorus, and Oestrogen Levels in OVX Rats

Compared to the sham group, serum calcium, phosphorus, and oestrogen levels in OVX rats were significantly ($p \leq 0.05$) lowered by 32.27%, 77.4%, and 30.95%, respectively (Figure 2). The data indicate that feeding control *dahi* did not improve the serum calcium levels of OVX rats ($p > 0.05$). In contrast, SDG-enriched *dahi* elevated blood calcium levels in OVX rats by 37.37% and was comparable to the sham group. Feeding control *dahi* improved these levels significantly ($p < 0.05$), but more improvement was observed in the SDG-enriched group. As expected, ovariectomy reduced serum oestrogen from 42 pg/mL (sham) to 29 pg/mL (OVX). Feeding control *dahi* raised serum oestrogen to 40.2 pg/mL, while SDG-enriched *dahi* raised it to 53.8 pg/mL.

Oestrogen deficiency is associated with a loss of serum calcium and phosphorus through the impairment of intestinal re-absorption. A 1.47- and 1.51-fold drop in serum calcium and phosphorus after ovariectomy was reported [24]. PEP raised serum calcium and phosphorus 1.20- and 1.27-fold in OVX rats. Another study demonstrated that caffeine significantly elevated serum calcium levels in OVX rats compared to the sham group. However, caffeine did not significantly raise the serum phosphorus and oestrogen levels of OVX rats [33]. Ground flaxseed with glucocorticoids raised calcium and phosphorus serum levels in female rats with induced bone loss compared to those in the positive control group [34]. Rutin restored serum oestrogen levels in OVX rats [32]. Similarly, serum oestrogen levels dropped significantly after ovariectomy, whereas treatment with Pu-erh tea extract in large doses restored the oestrogen. Unlike our findings, they detected no difference in serum calcium and phosphorus levels between OVX animals and control [35].

### 3.5. Effect of SDG-Enriched Dahi on Tibial Anthropometric Parameters and Bone Ash Calcium in OVX Rats

The impact of SDG-enriched *dahi* on the tibial anthropometric parameters of OVX and sham-operated rats was evaluated (Table 2). A significant difference in the tibia length among the experimental groups eight weeks post ovariectomy was not found ($p > 0.05$), while, in comparison to the sham group, significant reductions ($p \leq 0.05$) were observed in the tibial dry weight (28.59%), ash weight (24%), and bone ash calcium levels (48.46%) (Figure 2) in rats subjected to OVX. The administration of control *dahi* resulted in

a significant improvement ($p \leq 0.05$) in the ash weight, dry weight, and bone ash calcium compared to those in the OVX group but was significantly lower ($p \leq 0.05$) when compared to those in the sham group. In contrast, the administration of the SDG-enriched *dahi* restored the ash weight, dry weight, and bone ash calcium and was comparable to those in the sham control. These findings support previous research showing that OVX reduced bone density, ash weight, calcium, and phosphorus, while administering a combination of herbs and the polar fraction of *Punica granatum* L. peel extract reversed these effects [36,37]. Similarly, in rats subjected to OVX, femora dry weight (34.64%), ash weight (41.40%), and bone ash calcium (40.79) decreased significantly compared to those in the sham group. Compared to untreated rats, PEP from casein protected the loss in OVX rats' femora dry weight, ash weight, and bone ash calcium levels [24].

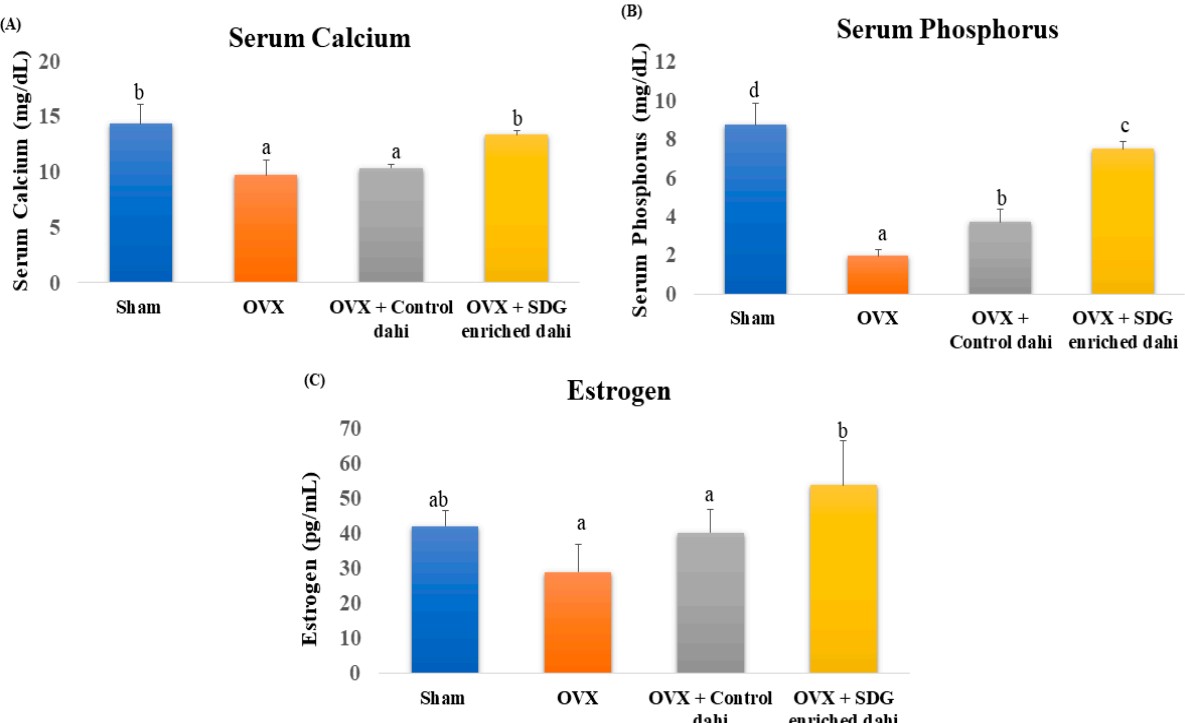

**Figure 2.** Effect of SDG-enriched set *dahi* on serum calcium, phosphorus, and oestrogen in OVX rats. (**A**) Serum calcium level; (**B**) serum phosphorus level; (**C**) serum oestrogen level. [abcd]: mean values within a column and within a parameter with at least one similar superscript do not differ significantly ($p > 0.05$). Values expressed as mean $\pm$ S.D. ($n = 5$).

*3.6. Effect of SDG-Enriched Set Dahi Administration on Tibial and Femoral BMD and Trabecular Microarchitecture in OVX Rats*

Three-dimensional images of the excised proximal tibial metaphysis region indicated trabecular bone loss in OVX animals compared to the sham group (Figure 3). The untreated OVX group showed a significant ($p \leq 0.05$) decrease in BMD (11.06%), BV/TV (4.46%), Tb. Th (12.66%), and Tb. N (57.83%) and a significant increase ($p \leq 0.05$) in Tb. Sp (90.69%) compared to the sham group (Figure 4). OVX rats treated with control and SDG-enriched *dahi* showed significant ($p \leq 0.05$) increases in BMD by 8.85% and 13.84%, respectively. Interestingly, the administration of SDG-enriched *dahi* improved the BMD by 3.13% compared to that in the sham group. Similarly, compared with OVX and sham-operated rats, SDG-enriched-*dahi*-fed ovariectomized rats had a 7.63% and 3.32% rise in BV/TV ratios. Trabecular thickness increased by 26.10% in SDG-administered OVX rats but not in control *dahi*-fed ovariectomized rats. The Trabecular number improved by 24.69% in control *dahi*-fed OVX rats, while SDG-enriched *dahi* increased the Tb. N by 57.94%, comparable to that in the sham control group.

Trabecular separation was reduced by 22.13% in control *dahi*-fed OVX rats and 88.82% in SDG-enriched-*dahi*-fed rats.

**Table 2.** Effect of SDG-enriched *dahi* on tibial anthropometric parameters and bone ash calcium in OVX rats.

| Group | Length (mm) | Ash Weight (mg) | Dry Weight (mg) | Calcium (mg/g of ash) |
|---|---|---|---|---|
| Sham | 30.96 ± 1.13 [a] | 264.53 ± 15.25 [c] | 604.65 ± 15.60 [c] | 176.62 ± 9.66 [c] |
| OVX | 31.02 ± 1.23 [a] | 201.02 ± 27.66 [a] | 431.78 ± 14.67 [a] | 91.03 ± 16.20 [a] |
| OVX and Control *dahi* | 30.87 ± 0.85 [a] | 232.38 ± 19.38 [b] | 492.12 ± 44.06 [b] | 131.60 ± 13.12 [b] |
| OVX and SDG-enriched *dahi* | 30.59 ± 1.09 [a] | 279.45 ± 14.41 [c] | 608.63 ± 24.26 [c] | 165.98 ± 4.68 [c] |

[abc]: mean values within a column and within an attribute with similar superscripts do not differ significantly ($p > 0.05$). Mean ± S.D. ($n = 6$).

Three-dimensional pictures of the excised distal femur epiphysis region indicated trabecular bone loss in OVX animals compared to the sham group (Figure 3). The micro-CT data analysis showed that the untreated OVX group had a significant ($p \leq 0.05$) decrease in BMD (9.18%), BV/TV (2.10%), Tb.Th (11.86%), and Tb.N (57.41%) compared to the sham group. The untreated OVX group also had increased Tb.Sp (87.70%) (Figure 5). OVX rats treated with control and SDG-enriched *dahi* showed a substantial increase in BMD by 5.77% and 12.22%, respectively ($p \leq 0.05$). The BV/TV ratio of control *dahi*-fed OVX rats decreased significantly ($p \leq 0.05$) by 7.20 percent relative to that of OVX rats, while that of SDG-enriched-*dahi*-fed rats increased significantly ($p \leq 0.05$). Trabecular thickness increased significantly ($p \leq 0.05$) in ovariectomized rats given control and SDG-enriched *dahi*. OVX rats fed SDG-enriched *dahi* showed 5.42 percent higher Tb.Th values than sham-operated rats. The trabecular number of OVX rats fed control *dahi* increased by 25.38 percent. However, the Tb.N. of OVX rats fed SDG-enriched *dahi* increased by 140.99 percent and 2.64 percent compared to those in OVX and sham-operated animals, respectively. Trabecular separation decreased by 18.89% ($p \leq 0.05$) in OVX rats fed control *dahi* and 47.22% in those fed SDG-enriched *dahi*.

OVX rats lose trabecular bone like postmenopausal women due to oestrogen deprivation. The microarchitecture of trabecular bone predicts OVX-induced bone loss and quality decline. In oestrogen-deficient (OVX) rats, SDG-enriched *dahi* improved trabecular bone microarchitectures and may be helpful for osteoporosis. This work confirms previous micro-CT findings that normal trabecular bone microarchitecture deteriorates significantly following ovariectomy [24,38]. Administering PEP (50–100 μg/Kg) improved BMD, BV/TV, Tb. N., and Tb. Th. and lowered Tb. Sp. levels in the femur and tibia [24]. *Bifidobacterium longum* supplementation enhanced BMD, BV/TV, and Tb.Th while lowering Tb. Sp levels in ovariectomized mice [38]. The administration of *Lactiplantibacillus plantarum* MGE 3038 to OVX female Wistar rats showed increases in BMD, trabecular Bone Volume, Tb. N., and Tb. Th and a decrease in trabecular spacing compared with the OVX group [14]. Similarly, *Lactiplantibacillus plantarum* GMNL 662 was reported to attenuate postmenopausal osteoporosis in OVX rats through the modulation of proinflammatory cytokines and bone metabolism-related markers [10]. Another study on the effect of probiotic *Lactobacillus reuteri* in preventing postmenopausal bone loss showed that oral administration of *L. reuteri* ($1 \times 10^9$ cfu/mL) increased femur distal trabecular BMD, Tb. Th., Tb.N and BV/TV by 24.02%, 15.36%, 48.95%, and 73.53%, respectively, while Tb. Sp. was decreased by 61.38% compared to those in OVX mice [16]. In a similar study, *Lactobacillus acidophilus* was administered orally ($10^9$ cfu/mL) to ovariectomized mice. The study reported that compared to OVX mice the BV/TV, Tb. Th., and Tb. N., in the femur trabecule was increased by 92.35%, 95.95%, and 104.29%, respectively, while the Tb. Sp. was decreased by 106.45%. Similarly, the BV/TV, Tb. Th., and Tb. N. of the tibia trabecule was increased by 25.86%, 60%, and 41.2%, respectively, while the Tb. Sp. was decreased by 60.48% [15]. These studies suggest that certain probiotic strains of Lactobacillus increase the bioavailability of

calcium by producing short-chain fatty acids (SCFAs) that lower gut pH, making it easier for calcium to dissolve and be absorbed; further, they exhibit immunomodulatory effects on the host immune system [17]. A synbiotic formulation composed of *Lactiplantibacillus plantarum*, *Levilactobacillus brevis*, *Leuconostoc mesenteroides*, *Pseudomonas fluorescens*, and *Pichia kudriavzevii* together with prebiotic dietary fibres was proved to reduce OVX-induced trabecular bone loss in mice through the production of bone-protective SCFAs, vitamin K2, and branched chain amino acids [11].

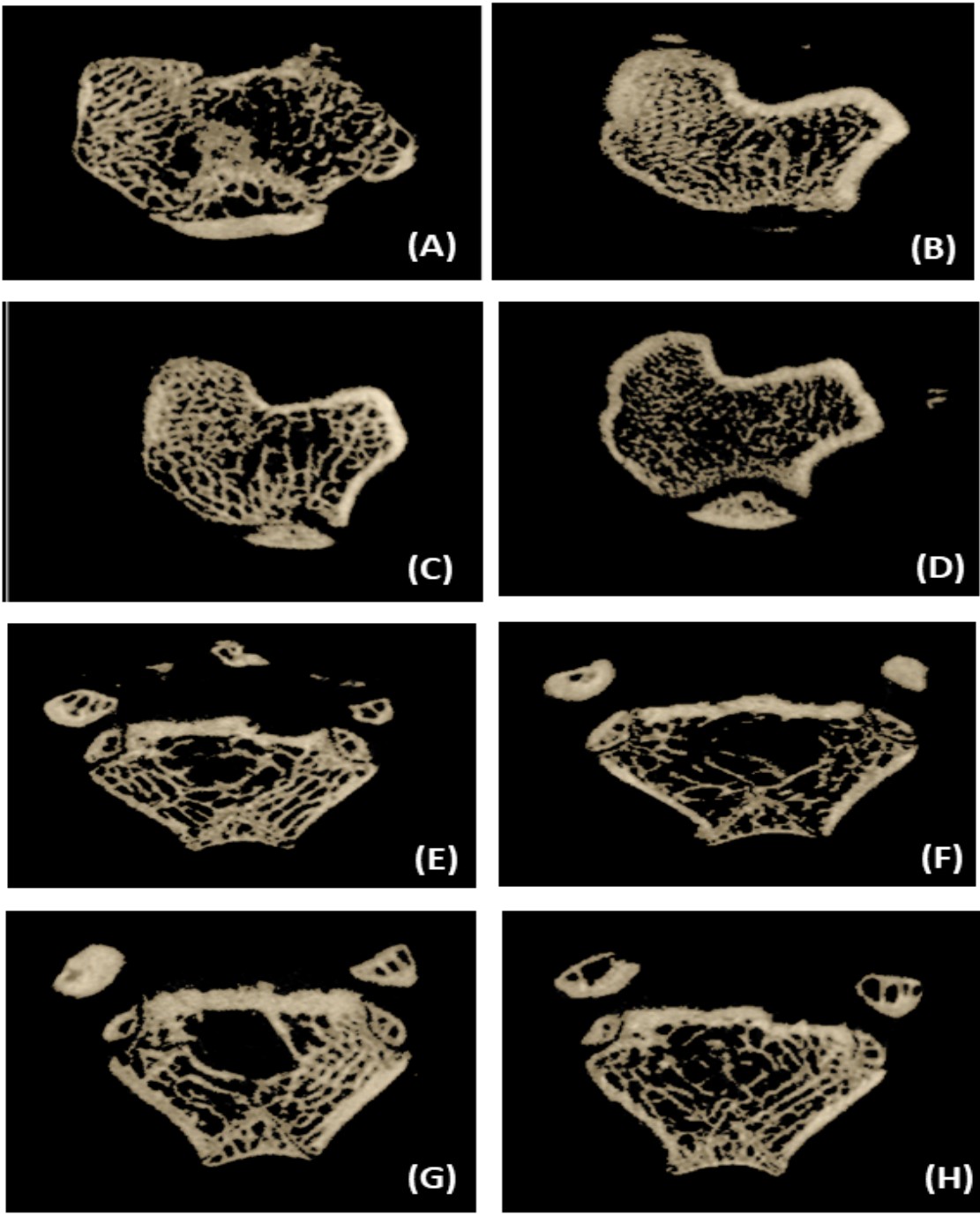

**Figure 3.** Representative micro-CT 3-D images of proximal tibial metaphysis and distal femoral epiphysis regions. (**A**) Tibia trabecule—sham control. (**B**) Tibia trabecule—OVX control. (**C**) Tibia trabecule—OVX and control *dahi*. (**D**) Tibia trabecule—OVX and SDG-enriched *dahi*. (**E**) Femur trabecule—sham control. (**F**) Femur trabecule—OVX control. (**G**) Femur trabecule—OVX and control *dahi*. (**H**) Femur trabecule—OVX and SDG-enriched *dahi*.

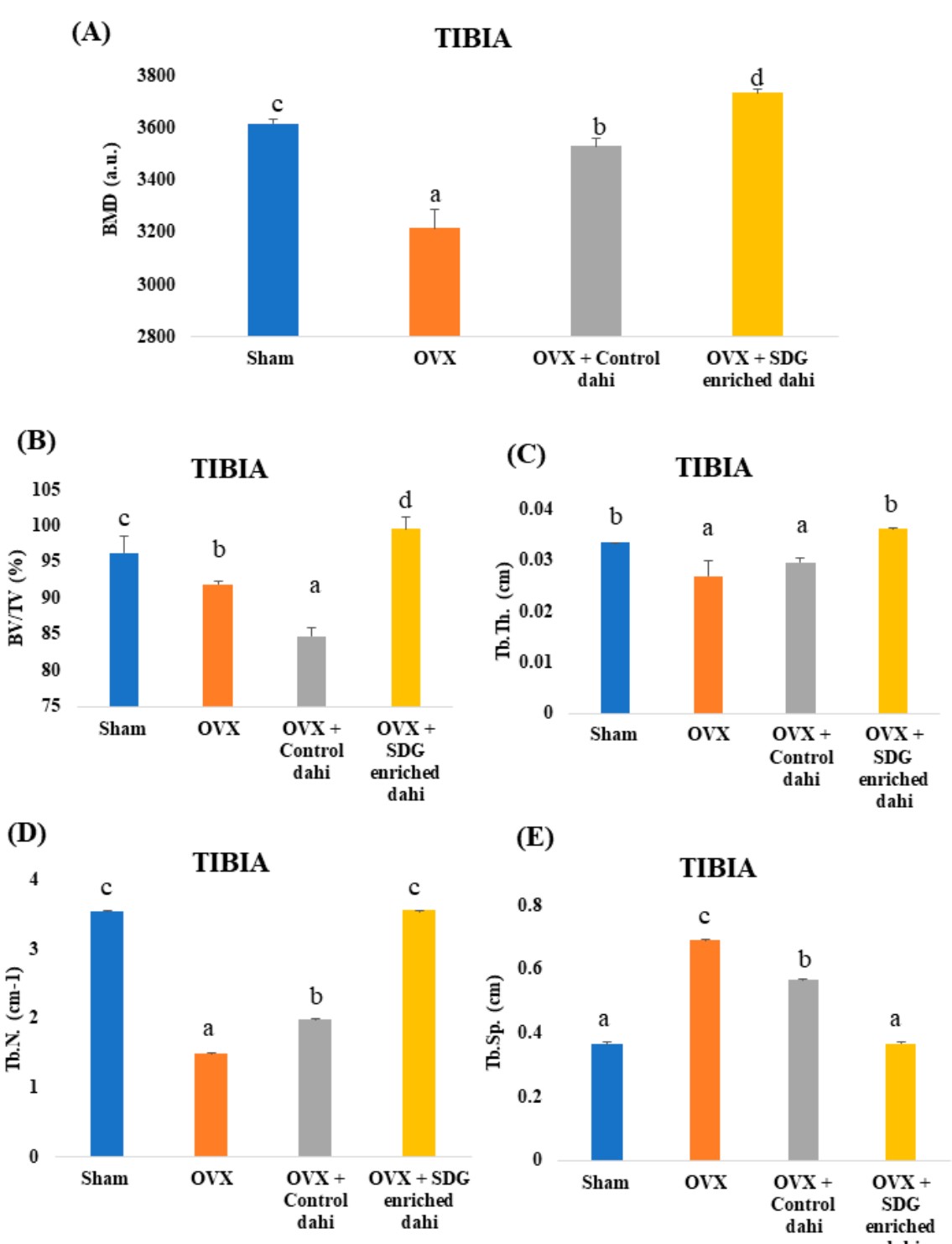

**Figure 4.** Effect of SDG-enriched set *dahi* administration on tibial BMD and trabecular microarchitecture in OVX rats. (**A**) BMD; (**B**) BV/TV ratio; (**C**) Trabecular thickness; (**D**) Trabecular number; (**E**) Trabecular separation. [abcd]: mean values within a column and within an attribute with similar superscripts do not differ significantly ($p > 0.05$). Mean ± S.D. ($n = 6$), BMD—bone mineral density, BV—Bone Volume, TV—Tissue Volume, a.u.—Arbitrary Units.

Similarly, chlorogenic acid at 45 mg/kg/d improved bone quality by modifying BMD (100%) and trabecular microarchitecture (Tb. N—85.19%, Tb. Th—100%, BV/TV—166.67%) in OVX rats, compared to the sham group [39]. OVX mice had lower BMD, trabecular BV/TV, Tb. Th, Tb. N, and Tb. Sp than sham-operated mice six weeks after surgery, but

17 g epoxyeicosatrienoic acid/kg 6 times per week for six weeks prevented trabecular bone loss [40]. Comparable results with regulatory B cells and ovariectomy-induced bone loss were also reported [41].

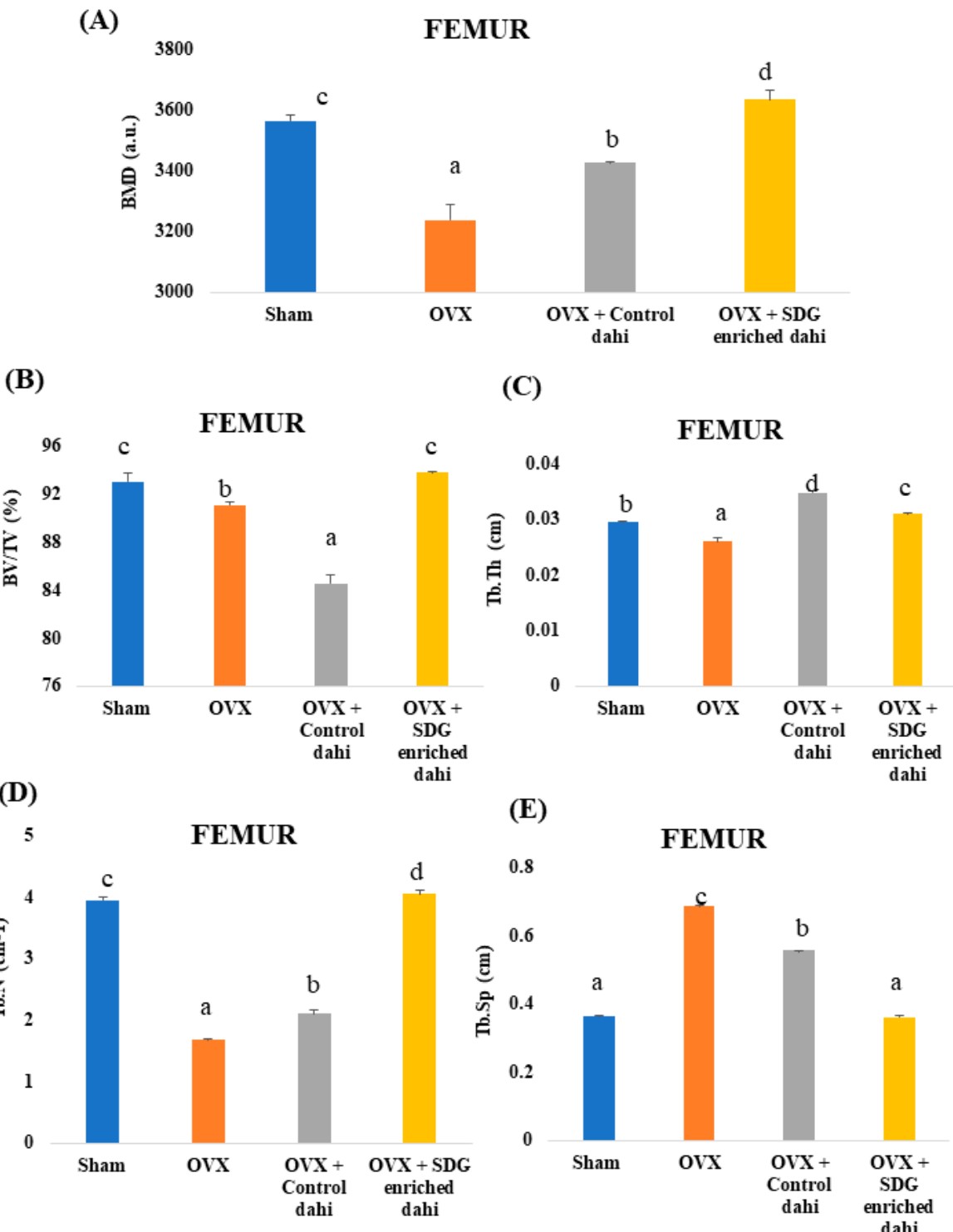

**Figure 5.** Effect of SDG-enriched set *dahi* administration on femoral BMD and trabecular microarchitecture in OVX rats. (**A**) BMD; (**B**) BV/TV ratio; (**C**) Trabecular thickness; (**D**) Trabecular number; (**E**) Trabecular separation. [abcd]: mean values within a column and within an attribute with similar superscripts do not differ significantly ($p > 0.05$). Mean ± S.D. ($n = 6$), BMD—Bone Mineral Density, BV—Bone Volume, TV—Tissue Volume, a.u.—Arbitrary Units.

The effect of flaxseed oil (11% fat by weight) on sham-operated and OVX rats was analysed for 12 weeks. OVX rats fed flaxseed oil had a 4.51% and 5.96% decrease in BV/TV and Tb. N compared to sham rats, while the Tb. Th was improved by 4.17%. Flaxseed oil boosted Tb. Sp levels by 17.34% in OVX rats compared to the control [42]. Another study found a decrease in the bone mineral density (BMD) by 28.13 percent in rats with glucocorticoid-induced bone loss (positive control) compared to rats fed a regular diet. Bone loss-induced rats fed 3%, 5%, and 7% powdered flaxseed had BMD increases of 1.08%, 22.69%, and 23.97% compared to the positive control group [34]. Osteoporosis is considered an inflammatory disorder. Probiotics can improve bone health since they are associated with several immunomodulatory activities. Also, probiotics maintain the balance between bone formation by osteoblasts and bone resorption by osteoclasts [17]. But postmenopausal osteoporosis occurs due to the deficiency of oestrogen. The current investigation clearly shows that the *dahi* enriched with probiotic (Group 3) improved the BMD and trabecular microarchitecture of both the femur and tibia compared to OVX control (Group 2), while the administration of SDG in the presence of the probiotic *Lactiplantibacillus plantarum* A5 (Group 4) improved the BMD and trabecular microarchitecture of both the femur and tibia better than the probiotic alone. The probiotic organism might have enhanced the conversion of SDG to mammalian lignan END and ENL, thereby improving their bioavailability to the oestrogen receptors. The mechanism behind the synergestic role of *Lactiplantibacillus plantarum* A5 with SDG needs to be explored.

### 3.7. Effect of SDG-Enriched Set Dahi on Serum Level of Bone Turnover Markers in OVX Rats

The uncoupling of bone formation and resorption caused by oestrogen deficiency increases the serum levels of bone turnover markers such as alkaline phosphatase (ALP) and osteocalcin (OCN) at the sites of bone remodelling. Therefore, it would be interesting to investigate the effect of SDG-enriched *dahi* on bone turnover markers in OVX rats. As shown in Figure 6, OVX significantly ($p \leq 0.05$) enhanced ALP activity (83.99%) and OCN levels (136.04%) compared to those in sham-operated animals. Control *dahi* treatment in OVX rats for eight weeks significantly ($p \leq 0.05$) reduced the bone remodelling rate by 16.85% and the OCN level by 37.83%, while SDG-enriched *dahi* administration significantly ($p \leq 0.05$) reduced ALP activity by 44.57% compared to that in OVX rats and 33.33% compared to that in control *dahi*-fed OVX rats. In rats fed SDG-enriched *dahi*, OCN levels reduced by 55.89% compared to those in OVX animals and 29.05% compared to those in control *dahi*-fed OVX rats ($p \leq 0.05$).

The biochemical metabolites of bone turnover detect osteoporosis severity and progression. ALP is an osteoblast-produced enzyme that calcifies bone matrix. ALP levels are high in postmenopausal osteoporosis, which causes excessive bone turnover and rapid bone remodelling [24]. OCN is essential to bone mineralization due to its calcium affinity. Deficits in calcium and phosphorus can raise circulatory OCN levels in osteoporosis. As expected, the OVX group had much greater ALP and OCN serum levels than the sham group. Interestingly, SDG-enriched *dahi* ameliorated the serum ALP and OCN elevations induced by OVX more effectively than control *dahi*. Elevated levels of ALP and OCN in ovariectomized rats were reported earlier [43]. But, in contrast to our results, they observed a significant increase in alkaline phosphatase following treatment with flaxseed extract. Increases of 2.26-fold and 1.65-fold in ALP and OCN in OVX rats compared to sham-operated rats were reported, while PEP (50 and 100 µg/kg) administration for eight weeks lowered ALP and OCN levels in OVX rats by 1.57 and 1.05 times, respectively [24]. Similarly, there was a significant increase in ALP and OCN in OVX rats, whereas 30 g/kg b.w of mixed herbs (sage, rosemary, and thyme) for one month decreased ALP and OCN levels in OVX rats [36].

Based on the findings of the rat study, it can be deduced that a dosage of 260 mg of SDG may potentially treat postmenopausal osteoporosis. On conversion of the animal dose to a human-equivalent dose (HED) [44], it was found that the initial dose for clinical

studies could be 1163.76 mg SDG/60 kg women/day (1.16 g SDG/ 60 kg women/day) (Refer to the Supplementary File for HED calculation). However, in order to facilitate the commercialization of the developed product, it is important to perform toxicological studies and clinical trials.

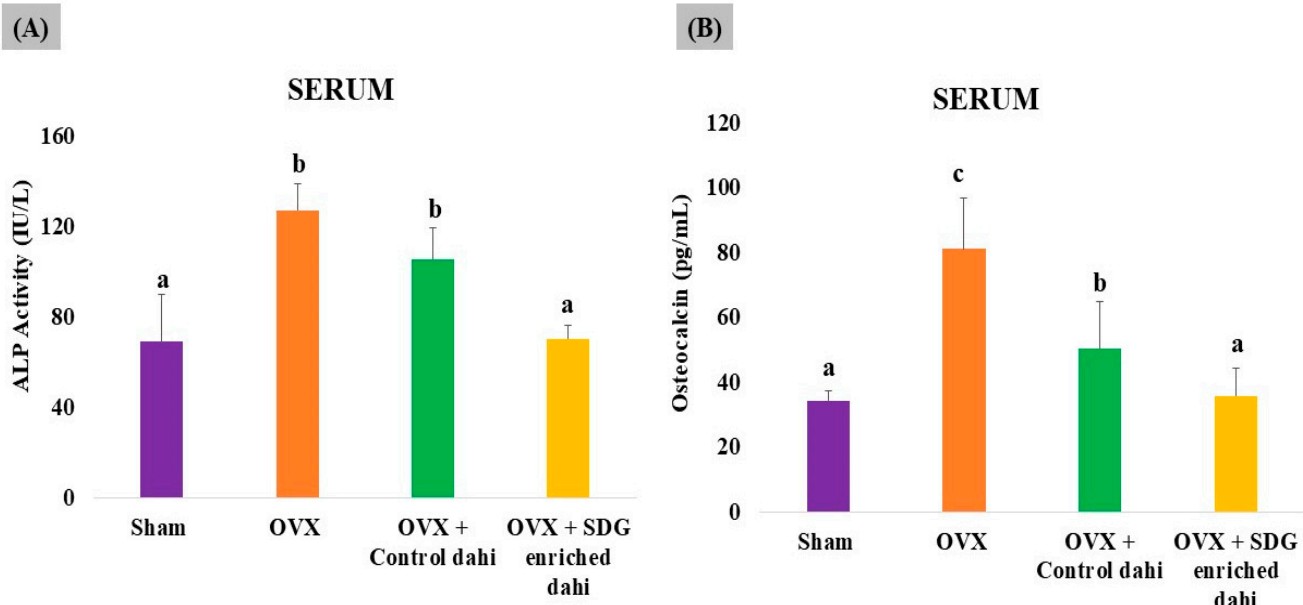

**Figure 6.** Effect of SDG-enriched set *dahi* on alkaline phosphatase activity and osteocalcin levels in OVX rats. (**A**) Alkaline phosphatase activity; (**B**) osteocalcin levels. [abc]: mean values within a column and within an attribute with at least one similar superscript do not differ significantly ($p > 0.05$). Mean ± S.D. ($n = 6$).

## 4. Conclusions

The present study illustrated that SDG-enriched fermented milk exhibits antiosteopenic effects in ovariectomized rats. Currently, there are no foods or dairy products available in the commercial market exclusively for postmenopausal women based on phytoestrogens. According to recent statistics given by the WHO, 30 percent of postmenopausal women suffer from osteoporosis. It has been reported that 61 percent of people in India have osteoporosis and, out of these, 80 percent are women. Since *dahi* or similar fermented dairy products are widely consumed by people of all ages, especially the geriatric population, enriching them with functional ingredients such as SDG will improve the therapeutic and functional value of the product. With the provision of using flaxseed lignan as a nutraceutical as per FSS (Health Supplements-Nutraceuticals) Regulations (2022), there is a scope for new functional fermented milk development to improve the quality of life of menopausal women.

**Supplementary Materials:** The following supporting information can be downloaded at https: //www.mdpi.com/article/10.3390/fermentation10020072/s1, 1. TA settings and parameters used for the texture analysis of set *dahi* with the back extrusion method. 2. Procedure for the measurement of serum calcium levels. 3. Procedure for the measurement of serum phosphorus levels. 4. Standard curve used for the estimation of osteocalcin. 5. Conversion of animal dose to human dose.

**Author Contributions:** E.T.: data curation formal analysis; investigation ; methodology; visualization; writing—original draft; writing—review and editing. N.R.P.: conceptualization; data curation; formal analysis; funding acquisition; project administration; resources; visualization; writing—review and editing. S.G.: conceptualization; funding acquisition; methodology; project administration; resources. S.D.: investigation. S.K.: resources; writing—review and editing. A.K.S.: supervision. All authors have read and agreed to the published version of the manuscript.

**Funding:** This research was funded by the Department of Science and Technology, Ministry of Science and Technology, Government of India (DST/SEED/WS/58) and Director, ICAR- National Dairy Research Institute, Karnal, India in the form of Institutional Fellowship. "The APC was funded by Department of Science and Technology, Ministry of Science and Technology, Government of India (DST/SEED/WS/58)".

**Institutional Review Board Statement:** The present study was conducted with approval from the Institutional Animal Ethics Committee (IAEC, 1705/GO/ac/13/CPCSEA Dt. 3 July 2013), ICAR-National Dairy Research Institute, Karnal, India.

**Informed Consent Statement:** Not applicable.

**Data Availability Statement:** The data that support the findings of this study are available on request from the corresponding author. The data are not publicly available due to privacy and lack of prior approval from the funding agency.

**Conflicts of Interest:** The authors declare no conflicts of interest.

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
