# Peer review of "Development and Validation of Flaxseed Lignan-Enriched Set-Type Fermented Milk to Manage Postmenopausal Osteoporosis"

_fermentation, doi:10.3390/fermentation10020072_

Round 1
Reviewer 1 Report
Comments and Suggestions for Authors
The experiment used dahi with added probiotics, prepared to contain 260 mg of SDG in 20 g of dahi. Three-month-old female Albino Wistar rats were ovariectomized (OVX) and divided into three groups: OVX control, OVX + dahi control, and OVX + SDG-enriched dahi. The study showed that after ovariectomy, the concentration of calcium in rats' serum, estrogen and calcium in bone ashes decreased compared to the control group. The authors show that daily administration of SDG-with dahi for eight weeks restored normal parameters. Examination of the proximal tibial metaphysis and distal femoral epiphysis showed that ovariectomy reduced bone mineral density by 11.06% and 9.18%, respectively, and trabecular thickness (Tb. Th) by 12.66% and 28 11.86% respectively, increased trabecular separation (Tb.Sp.) by 90.69% and 87.70%, respectively, compared to sham control rats. Dahi supplemented with SDG improved BMD by 16.06 and 30 12.24% and Tb.Th by 35.32 and 19.62%, respectively, and reduced Tb.Sp by 47.04 and 47.22%, respectively, in OVX rats. The results suggest that the developed dahi kit may help in the treatment of postmenopausal osteoporosis.
The work is written correctly, the experiments were planned and carried out correctly. The animals were well taken care of. It's not entirely clear to me how many measurements were taken, the authors wrote: "The pH values ​​were measured twice at three different times to monitor fermentation." (117-118)
Providing more precise data would improve the substantive value of the work.
The description of the methods is correct and the analyzes are reliable.
Correct discussion. In the conclusions, the authors somehow suggest that there is no special line of exclusive dairy products for postmenopausal women, the question is whether it is needed...
Author Response
The suggestion of reviewer 1 has been addressed

Reviewer 2 Report
Comments and Suggestions for Authors This work showsts the results of a study on the effect of a bacteria, with probiotic characteristics added to fermented milk, on bone development for its potential use in human osteoporosis. The introduction does not mention some works closely related to this and that should be part of its theoretical background. It is requested to review recent related publications to complete the introduction. The materials and methods are adequate except that a general observation of the state of the organs during the necropsy is not proposed; This, added to the taking of samples for histological studies of key organs such as the liver and kidney, would be important to recognize toxic effects. If not, the knowledge we have about the toxicity of the substance produced should be clearly stated in the text. In the results and discussion section; The results are well described but the discussion is very superficial. Perhaps this is because, as mentioned, the bibliography used is incomplete and many works are missing (e.g. Jin et al., Lawenius et al) that are very important to discuss with the results of the present study, even some of these works use the same bacteria. The figures are good although the Microtomography photographs must be improved General comment: The work focuses on the beneficial effects on bone tissue. However, the potential effects on the female reproductive system are not considered. On this system, phytoestrogens have been shown to have both beneficial and harmful actions; for example, they have been related to the pathogenesis of some forms of cancer and endometriosis. Although a study of this system could be a good topic for another paper, it should at least be considered that it is a fundamental aspect to take into account considering that these negative effects on the genital organs are undervalued in the references. It would be very important to propose a future study with non-ovaryhysterectomized females to corroborate. Specific comment: Line 176 Change the term autopsied by the more adequate necropsied
Author Response
The suggestion of Reviwer 2 has been addressed

Round 2
Reviewer 2 Report
Comments and Suggestions for Authors With the corrections made, the manuscript is ready to be published.